# TOWARDS MORE ACCURATE DIFFUSION MODEL ACCELERATION WITH A TIMESTEP ALIGNER

## ABSTRACT

A diffusion model, which is formulated to produce an image using thousands of denoising steps, usually suffers from a slow inference speed. Existing acceleration algorithms simplify the sampling by skipping most steps yet exhibit considerable performance degradation. By viewing the generation of diffusion models as a discretized integrating process, we argue that the quality drop is partly caused by applying an inaccurate integral direction to a timestep interval. To rectify this issue, we propose a **timestep aligner** that helps find a more accurate integral direction for a particular interval at the minimum cost. Specifically, at each denoising step, we replace the original parameterization by conditioning the network on a new timestep, which is obtained by aligning the sampling distribution to the real distribution. Extensive experiments show that our plug-in design can be trained efficiently and boost the inference performance of various state-of-the-art acceleration methods, especially when there are few denoising steps. For example, when using 10 denoising steps on the popular LSUN Bedroom dataset, we improve the FID of DDIM from 9.65 to 6.07, simply by adopting our method for a more appropriate set of timesteps. Code will be made publicly available.

## 1 INTRODUCTION

Diffusion probabilistic models (DPMs) (Sohl-Dickstein et al., 2015; Ho et al., 2020; Song et al., 2020), known simply as diffusion models, have recently received growing attention due to its efficacy of modeling complex data distributions (Ho et al., 2020; Nichol & Dhariwal, 2021; Dhariwal & Nichol, 2021). A DPM first defines a forward diffusion process (*i.e.*, either discrete-time (Ho et al., 2020; Song et al., 2021) or continuous-time (Song et al., 2020)) by gradually adding noise to data samples, and then learns the reverse denoising process with a timestep-conditioned parameterization. Consequently, it usually requires thousands of denoising steps to synthesize an image, which is time-consuming (Ho et al., 2020; Song et al., 2021; Kong & Ping, 2021).

To accelerate the generation process of diffusion models, a common practice is to reduce the number of inference steps. For example, instead of a step-by-step evolution from the state of timestep 1,000 to the state of timestep 900, previous works (Song et al., 2020; Ho et al., 2020; Bao et al., 2022) manage to directly link these two states with a one-time transition. That way, it only needs to evaluate the denoising network once instead of 100 times, thus substantially saving the computational cost.

A side effect of the above acceleration pipeline is the performance degradation that appears as artifacts in the synthesized images. In this paper, we aim to identify and address the cause of this side effect. As the gray dashed line shown in Fig. 1a, the generation process of diffusion models can be viewed as a discretized integrating process, where the direction of each integral step is calculated by the pre-learned noise prediction model. To reduce the number of steps, existing algorithms (Song et al., 2021; Bao et al., 2022; Nichol & Dhariwal, 2021) typically apply the direction predicted for the initial state for the following timestep interval, as the red line shown in Fig. 1a, resulting in a gap between the sampling distribution and the real distribution. Karras (Karras et al., 2022) identify this distribution gap as the *truncation error*, which accumulates over the whole steps intuitively and theoretically. As the red line shown in Fig. 1b, the gap between the sampling distribution and the real distribution increases as the integrating process evolves.

To alleviate the problem arising from skipping steps, we propose a timestep aligner, termed as `DPM-Aligner`, which targets at finding a more accurate integral direction for a particular interval.

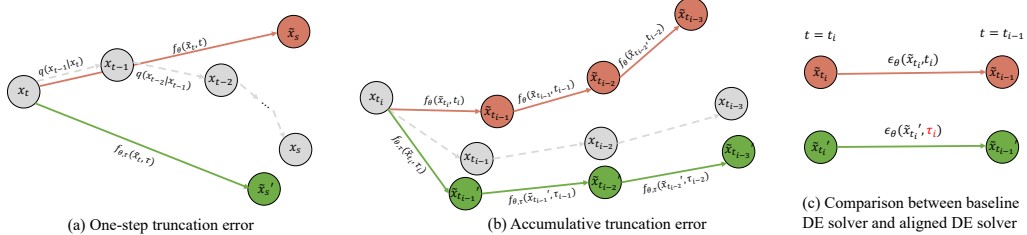

Figure 1: **Conceptual description** of (a) the one-step truncation error, (b) the accumulative truncation error, and (c) re-aligning the sampling and the real distribution of our DPM-Aligner by replacing the input timestep from $t$ to $\tau$, by the full-step reverse process (**gray dashed line**), the baseline acceleration pipeline (**red line**), and our proposed method with timestep alignment (**green line**).

As the green line shown in Fig. 1a, our approach is designed to achieve this purpose efficiently by searching a more appropriate timestep $\tau$ replacing the previous $t$ as the new condition input of the pre-learned noise prediction model. By doing so, we are able to significantly reduce the one-step truncation error, and hence the accumulative truncation error, as demonstrated with the green line in Fig. 1a and Fig. 1b respectively. Our motivation is intuitive – it is based on the observation that "*although the direction estimated from the initial state may not be appropriate for integration on the interval, the one from some intermediate state can be*", akin to the mean value theorem of integrals. To obtain this new timestep, we re-align the sampling distribution to the real distribution by optimizing a specially designed loss function. We theoretically prove the feasibility of DPM-Aligner, and provide an estimation of the error bound for deterministic sampling algorithms. Experiments using different numbers of function evaluations (NFE) show that our DPM-Aligner can be used to boost the sampling quality of various acceleration methods without extra time cost, (*e.g.*, DDIM (Song et al., 2021), Analytic-DPM (Bao et al., 2022), DPM-solver (Lu et al., 2022), *etc.*) in a plug-in fashion. Hence, our work offers a new perspective on accelerating the inference while simultaneously reducing the quality degradation of diffusion models.

## 2 RELATED WORK

**DPMs and the applications.** Diffusion probabilistic model (DPM) is initially introduced by Sohl-Dickstein *et al.* (Sohl-Dickstein et al., 2015), where the training is based on the optimization of the variational lower bound $L_{vb}$. Denoising diffusion probabilistic model (DDPM) (Ho et al., 2020) proposes a re-parameterization trick of DPM and learns the reweighted $L_{vb}$. Song *et al.* (Song et al., 2020) model the forward process as a stochastic differential equation and introduce continuous timesteps. With rapid advances in recent studies, DPMs show great potential in various downstream applications, including speech synthesis (Chen et al., 2020; Kong et al., 2020), video synthesis (Ho et al., 2022b), super-resolution (Saharia et al., 2021; Li et al., 2022), conditional generation (Choi et al., 2021), and image-to-image translation (Saharia et al., 2022; Sasaki et al., 2021).

**Faster DPMs** attempt to explore shorter trajectories rather than the complete reverse process, while ensuring the synthesis performance compared to the original DPM. Existing methods can be divided into two categories. The first category includes knowledge distillation (Salimans & Ho, 2022; Luhman & Luhman, 2021; Song et al., 2023). Although such methods may achieve respectable synthesis performance with only one-step generation (Song et al., 2023), they require expensive training stages before they are applied to efficient sampling, leading to poor applicability. The second category consists of training-free methods suitable for pre-trained DPMs. DDIM (Song et al., 2021) is the first attempt to accelerate the sampling process using a probability flow ODE (Song et al., 2020). Some existing methods search for the integration trajectories using grid search (Chen et al., 2020). However, it is only suitable for a short reverse process due to its exponentially growing time complexity. Some methods try to search for the trajectories by solving a least-cost-path problem with a dynamic programming (DP) algorithm or using the analytic solution (Watson et al., 2021; Bao et al., 2022). Another representative category of fast sampling methods use high-order differential equation (DE) solvers (Jolicoeur-Martineau et al., 2021; Liu et al., 2022; Popov et al., 2022; Tachibana et al., 2021; Lu et al., 2022). Saharia *et al.* (Saharia et al., 2021) and Ho *et al.* (Ho et al., 2022a) manage to

(a) DDIM sampler, NFE = 10
quadratic trajectory

(b) DDIM sampler, NFE = 20
quadratic trajectory

(c) DDIM sampler, NFE = 10
uniform trajectory

(d) DPM-Solver-2 sampler, NFE = 10
log-SNR trajectory

Figure 2: **Quantitative measurement** of the gap between the real and the sampling distribution using DDIM (Song et al., 2021) and DPM-Solver-2 (Lu et al., 2022). The horizontal axis represents the timesteps which form a (a) quadratic trajectory with NFE = 10; (b) quadratic trajectory with NFE = 20; (c) uniform trajectory with NFE = 10; (d) log-SNR trajectory with NFE = 10. We plot the $L_2$ distance between $\mathbf{x}_t$ and $\widetilde{\mathbf{x}}_t$ for the original and the aligned sampler, which are shown in red and blue, respectively. We also provide an error bound for deterministic sampler theoretically in Theorem 2.

train DPMs using continuous noise level and draw samples by applying a few-step discrete reverse process. Some GAN-based methods also consider larger sampling step sizes, e.g., in (Xiao et al., 2022) a multi-modal distribution is learned in a conditional GAN with a large step size. However, to the best of our knowledge, existing training-free acceleration algorithms are bottlenecked by the poor sampling performance with extremely few inference steps (*e.g.*, less than 5 steps). Our `DPM-Aligner` can be considered as a performance booster for existing training-free acceleration methods, *i.e.*, it further improves the generation performance.

## 3 METHOD

### 3.1 BACKGROUND

Suppose that $\mathbf{x}_0 \in \mathbb{R}^D$ is a $D$-dimensional random variable with an unknown distribution $q_0(\mathbf{x}_0)$. Diffusion probabilistic models (DPMs) (Sohl-Dickstein et al., 2015; Song et al., 2020; Ho et al., 2020) define a forward process $\{\mathbf{x}_t\}_{t \in [0,T]}$ by gradually adding noise onto $\mathbf{x}_0$ with $T > 0$, such that for any $t \in [0, T]$, we have the transition distribution:

$$q_{0t}(\mathbf{x}_t|\mathbf{x}_0) = \mathcal{N}(\mathbf{x}_t; \alpha_t \mathbf{x}_0, \sigma_t^2 \mathbf{I}), \tag{1}$$

where $\alpha_t, \sigma_t \in \mathbb{R}^+$ are differentiable functions of $t$ with bounded derivatives. The choice of $\alpha_t, \sigma_t$ is referred to as the *noise schedule*. Let $q_t(\mathbf{x}_t)$ be the marginal distribution of $\mathbf{x}_t$, DPM ensures that $q_T(\mathbf{x}_T) \approx \mathcal{N}(\mathbf{x}_T; \mathbf{0}, \sigma^2 \mathbf{I})$ for some $\sigma > 0$, and the signal-to-noise-ratio (SNR) $\alpha_t^2/\sigma_t^2$ is strictly decreasing w.r.t. the timestep $t$ (Kingma et al., 2021).

DPMs introduce a neural network $\boldsymbol{\epsilon}_\theta(\mathbf{x}_t, t)$, namely the noise prediction model, to approximate the score function from the given $\mathbf{x}_t$, where the parameter $\theta$ can be optimized by the objective below:

$$\mathbb{E}_{\mathbf{x}_0, \epsilon, t} \left[ \omega_t \|\boldsymbol{\epsilon}_\theta(\mathbf{x}_t, t) - \boldsymbol{\epsilon}\|_2^2 \right], \tag{2}$$

where $\omega_t$ is the weighting function, $\boldsymbol{\epsilon} \sim \mathcal{N}(\mathbf{0}, \mathbf{I})$, $\mathbf{x}_t = \alpha_t \mathbf{x}_0 + \sigma_t \boldsymbol{\epsilon}$, and $t \sim \mathcal{U}[0, T]$.

### 3.2 GAP BETWEEN REAL AND SAMPLING DISTRIBUTIONS

Note that a sampler of DPM builds upon the use of the noise prediction model $\boldsymbol{\epsilon}_\theta$ at each timestep $t$, which is technically a discretization of an integrating process. At each denoising step, one applies the noise prediction model $\boldsymbol{\epsilon}_\theta$ onto the intermediate result $\widetilde{\mathbf{x}}_t$ from the last denoising step together with its corresponding timestep condition $t$, *i.e.* $\widetilde{\boldsymbol{\epsilon}}_t = \boldsymbol{\epsilon}_\theta(\widetilde{\mathbf{x}}_t, t)$. The achieved noise prediction $\widetilde{\boldsymbol{\epsilon}}_t$ will be used as the integral direction towards the next denoised result.

However, recall that during the training of DPM, the noise prediction model $\boldsymbol{\epsilon}_\theta$ is trained with the noisy version of the real data at timestep $t$, *i.e.*, $\mathbf{x}_t = \alpha_t \mathbf{x}_0 + \sigma_t \boldsymbol{\epsilon}$, where $\boldsymbol{\epsilon} \sim \mathcal{N}(\mathbf{0}, \mathbf{I})$. Intuitively, due to the error of the DE solver (*i.e.*, SDE solver or ODE solver) using large discretization step sizes, there is a considerable gap between the distributions of the real distribution of $\mathbf{x}_t$ and the sampling distribution of $\widetilde{\mathbf{x}}_t$ at each timestep $t$, which is called the truncation error (Karras et al., 2022). Even

worse, this error is accumulated progressively during the reverse process, since the gap between the distributions of $\mathbf{x}_t$ and $\widetilde{\mathbf{x}}_t$ leads to a poor prediction of $\widetilde{\mathbf{x}}_{t-1}$, in which the truncation error at timestep $t$ transmits to the next timestep $t-1$, which is also claimed by Karras (Karras et al., 2022).

To support this insight, Fig. 1a shows the truncation error at each timestep, in comparison with the original full-step sampling process. As the gray dashed line in Fig. 1a indicates, in the original full-step sampling process, the denoised $\mathbf{x}_s$ comes from a step-by-step denoising refinement starting at $\mathbf{x}_t$, calculated by the results from the noise prediction model at all intermediate timesteps from $\mathbf{x}_t$ down to $\mathbf{x}_s$. However, the accelerated sampling algorithm (*i.e.*, the red line) uses a large step size and replaces all the intermediate predicted noises with one single predicted one at the initial timestep, *i.e.*, $\widehat{\boldsymbol{\epsilon}}_t = \boldsymbol{\epsilon}_\theta(\widetilde{\mathbf{x}}_t, t)$. Therefore, each $\widetilde{\mathbf{x}}_t$ incurs a truncation error due to a large discretization step sizes.

Furthermore, note that for a sampling process, given the value of $\widetilde{\mathbf{x}}_T$ at timestep $T$ and a timestep $t$ in $[0, T]$, the DE solver approximates the true $\mathbf{x}_t$ as $\widetilde{\mathbf{x}}_t$. As the red line indicates in Fig. 1b, since the local truncation error accumulates at each step, the gap between the sampling distribution of $\widetilde{\mathbf{x}}_t$ and the real distribution of $\mathbf{x}_t$ increases as $t$ evolves. To gain the insight of the accumulative error, we conduct a simple experiment to provide convincing evidence of the above observation using DDIM (Song et al., 2021) and DPM-Solver-2 (Lu et al., 2022) sampler. We first sample $\widetilde{\mathbf{x}}_T$ from CIFAR10 dataset (Krizhevsky & Hinton, 2009) and estimate the true $\mathbf{x}_t$ sequences using DDIM sampler with the number of function evaluations (NFE) $= 1,000$. Meanwhile, we draw the approximated $\widetilde{\mathbf{x}}_t$ sequences using DDIM and DPM-Solver-2 samplers following the quadratic, uniform and log-SNR trajectories with NFE $= 10$ or $20$, respectively. Then we calculate the $L_2$ metric of $\mathbf{x}_t - \widetilde{\mathbf{x}}_t$ at each timestep $t$, in order to demonstrate the gap of the two distributions, the result is shown in Fig. 2. It is noteworthy that: (1) the gap between $\mathbf{x}_t$ and $\widetilde{\mathbf{x}}_t$ indeed accumulates as timestep $t$ evolves from $T$ to 0, which indicates that the truncation error at each step makes the sampling distribution farther and farther away from the real one and severely hurts the final sampling quality; (2) with the same quadratic trajectory, the larger the NFE is, the smaller the gap between the real and sampling distributions is; (3) different types of trajectories and different DPM samplers account for different behaviors of the accumulative truncation error.

### 3.3 TIMESTEP ALIGNER FOR NOISE PREDICTION MODEL

Recall that in Sec. 3.2 we demonstrate the gap between the real and the sampling distribution. This gap will damage the quality of the synthesis samples. In this section, we propose the `DPM-Aligner` which re-aligns the distribution of approximated $\widetilde{\mathbf{x}}_t$ and the input condition timestep $t$. The solution and analysis are highly motivated by bridging the gap between the real distribution and sampling distribution at the same timestep $t$ using the mean value theorem of integrals.

Given the analysis that the sampling distribution of the predicted $\widetilde{\mathbf{x}}_t$ fails to follow the real marginal distribution $q_t(\mathbf{x}_t)$ at each timestep $t$. Therefore, what we need to do to boost any given acceleration algorihtm, is to choose an adequate timestep $\tau$, *replacing the input for the noise prediction model $\boldsymbol{\epsilon}_\theta$ from $t$ to $\tau$* (which is shown in Fig. 1c), such that the sampling distribution of $\widetilde{\mathbf{x}}_t$ tends to obey the distribution $q_t(\mathbf{x}_t)$. This greatly improves the poor performance of the noise prediction model $\boldsymbol{\epsilon}_\theta$, since $\boldsymbol{\epsilon}_\theta$ is trained using paired $(\mathbf{x}_t, t)$, in which $\mathbf{x}_t$ obeys the distribution $q_t(\mathbf{x}_t)$ at timestep $t$.

We first provide a picture of our formulation, as is shown in Fig. 1. In order to reduce the truncation error caused by inaccurate integral direction and large discretization step size (*i.e.*, the denoised result $\widetilde{\mathbf{x}}_s$ achieved along the red line in Fig. 1a), we target at finding a more accurate integral direction (*i.e.*, by replacing $t_i$ with re-aligned $\tau_i$ in Fig. 1c) for the interval from $\mathbf{x}_t$ to $\mathbf{x}_s$ (*i.e.*, the denoised result $\widetilde{\mathbf{x}}'_s$ achieved along the green line in Fig. 1a). In this sense, we are able to improve the accuracy of the noise prediction model and achieve a better integral direction for each interval (*i.e.*, the green line in Fig. 1b), mitigating the accumulation of the truncation error (*i.e.*, the red line in Fig. 1b).

Formally, for a discretization $0 = t_0 < t_1 < \cdots < t_K = T$ of $[0, T]$ and a DE solver $f_\theta(\mathbf{x}_{t_i}, t_i)$ which is responsible to denoise the intermediate $\widetilde{\mathbf{x}}_{t_i}$ for one single step, *i.e.*, $\widetilde{\mathbf{x}}_{t_{i-1}} = f_\theta(\widetilde{\mathbf{x}}_{t_i}, t_i)$. For instance, the DE solver $f_\theta$ of DDIM (Song et al., 2021) is of the following form:

$$f_\theta(\widetilde{\mathbf{x}}_{t_i}, t_i) = \frac{\alpha_{t_{i-1}}}{\alpha_{t_i}}\widetilde{\mathbf{x}}_{t_i} - \left(\frac{\alpha_{t_{i-1}}\sigma_{t_i}}{\alpha_{t_i}} - \sigma_{t_{i-1}}\right)\boldsymbol{\epsilon}_\theta(\widetilde{\mathbf{x}}_{t_i}, t_i). \tag{3}$$

---

**Algorithm 1** Training $\tau_i$

---

1: **for** $i = K, K - 1, \cdots, 1$ **do**
2:     **repeat**
3:         $\mathbf{x}_0 \sim q_0(\mathbf{x}_0), \boldsymbol{\epsilon} \sim \mathcal{N}(\mathbf{0}, \mathbf{I})$
4:         $\widetilde{\mathbf{x}}_T \leftarrow \alpha_T \mathbf{x}_0 + \sigma_T \boldsymbol{\epsilon}$
5:         $\widetilde{\mathbf{x}}_{t_i} \leftarrow f_{\theta,\tau}(f_{\theta,\tau}(\cdots (f_{\theta,\tau}(\widetilde{\mathbf{x}}_T, \tau_K) \cdots), \tau_{i+2}), \tau_{i+1})$
6:         Take gradient descent step on

$$\nabla_{\tau_i}(\|\boldsymbol{\epsilon}_\theta(f_{\theta,\tau}(\widetilde{\mathbf{x}}_{t_i}, \tau_i), t_{i-1}) - \boldsymbol{\epsilon}_\theta(\widetilde{\mathbf{x}}_{t_i}, t_i)\|_2^2)$$

7:     **until** converged
8: **end for**

---

`DPM-Aligner` will re-align $\widetilde{\mathbf{x}}_{t_i}$ and $t_i$, achieving the aligned DE solver $f_{\theta,\tau}$ by:

$$\widetilde{\mathbf{x}}'_{t_{i-1}} = f_{\theta,\tau}(\widetilde{\mathbf{x}}_{t_i}, \tau_i) := \frac{\alpha_{t_{i-1}}}{\alpha_{t_i}} \widetilde{\mathbf{x}}_{t_i} - \left( \frac{\alpha_{t_{i-1}} \sigma_{t_i}}{\alpha_{t_i}} - \sigma_{t_{i-1}} \right) \boldsymbol{\epsilon}_\theta(\widetilde{\mathbf{x}}_{t_i}, \tau_i), \tag{4}$$

where $\tau_i$ is the aligned timestep replacing the previous input condition timestep $t_i$.

Training `DPM-Aligner` is simple and efficient. Recall that we hope to find a timestep $\tau_i$ replacing the previous $t_i$, such that the sampling distribution of $\widetilde{\mathbf{x}}_{t_{i-1}} = f_{\theta,\tau}(\widetilde{\mathbf{x}}_{t_i}, \tau_i)$ tends to follow the real distribution $q_{t_{i-1}}(\mathbf{x}_{t_{i-1}})$. Therefore, after determining the number of function evaluations (NFE) $K$ and its corresponding trajectory $0 = t_0 < t_1 < \cdots < t_K = T$, the training process is to enforce the sampling distribution of intermediate $\widetilde{\mathbf{x}}_{t_i}$ and the real distribution $q_{t_i}(\mathbf{x}_{t_i})$ to coincide.

Concretely, we train the aligned timestep reversely from $t_K = T$ down to $t_1$. Given $\mathbf{x}_0 \sim q_0(\mathbf{x}_0)$ and $\boldsymbol{\epsilon} \sim \mathcal{N}(\mathbf{0}, \mathbf{I})$, the distribution of $\widetilde{\mathbf{x}}_T$ approximates the Gaussian distribution, *i.e.*, $\widetilde{\mathbf{x}}_T = \alpha_T \mathbf{x}_0 + \sigma_T \boldsymbol{\epsilon} \approx \mathcal{N}(\mathbf{0}, \sigma^2 \mathbf{I})$. For each $i = K, K - 1, \cdots, 0$, denote by $\widetilde{\mathbf{x}}_{t_i}$ the intermediate denoised result using the aligned DE solver $f_{\theta,\tau}$ in Eq. (4). The loss function of $\tau_i, i = K, K - 1, \cdots, 1$ is defined as:

$$\mathcal{L}_i(\tau_i) = \mathbb{E}_{\mathbf{x}_0, \boldsymbol{\epsilon}} \left[ \|\boldsymbol{\epsilon}_\theta(f_{\theta,\tau}(\widetilde{\mathbf{x}}_{t_i}, \tau_i), t_{i-1}) - \boldsymbol{\epsilon}_\theta(\widetilde{\mathbf{x}}_{t_i}, t_i)\|_2^2 \right]. \tag{5}$$

The training process is summarized in Algorithm 1. It is noteworthy that the training of our method is in a plug-in fashion. There is no need to modify the parameters of the pre-trained DPMs. Besides, since it only requires optimization of the $\tau_i$ through a denoising step, the training is extremely efficient (*e.g.*, `DPM-Aligner` takes $\sim 1$ hour in all of NFE = 10 case on an NVIDIA Tesla A100 GPU).

### 3.4 THEORETICAL FOUNDATIONS OF `DPM-ALIGNER`

We first exhibit the relationship between the objective of our method (*i.e.*, Eq. (5)) and that of the DDPM (*i.e.*, Eq. (2)). Formally, we have the following theorem. Proof is addressed in Appendix A.1.

**Theorem 1.** *Assume that $\boldsymbol{\epsilon}_\theta$ is the groundtruth noise prediction model. Then the training process of* `DPM-Aligner` *resembles that of the original DPM, i.e., for $i = K, K - 1, \cdots, 1$, the optimal $\tau_i = \arg\min_{\tau_i} \mathcal{L}_i(\tau_i)$ holds the following property:*

$$\arg\min_{\tau_i} \mathcal{L}_i(\tau_i) = \arg\min_{\tau_i} \mathbb{E}_{\mathbf{x}_0, \boldsymbol{\epsilon}} \left[ \|\boldsymbol{\epsilon}_\theta(f_{\theta,\tau}(\widetilde{\mathbf{x}}_{t_i}, \tau_i), t_{i-1}) - \frac{\widetilde{\mathbf{x}}_{t_i} - \alpha_{t_i} \mathbf{x}_0}{\sigma_{t_i}}\|_2^2 \right]. \tag{6}$$

We then give an error bound for the deterministic sampler, which derives the feasibility of the `DPM-Aligner` to re-align each timestep from $t_i$ to $\tau_i$ for a DPM. Proof is available in Appendix A.2.

**Theorem 2.** *Under the condition in Theorem 1, and let $f_{\theta,\tau}$ be a deterministic sampler. Assume that $\|\boldsymbol{\epsilon}_\theta(\mathbf{x}, t) - \boldsymbol{\epsilon}_\theta(\mathbf{y}, t)\|_2 \geqslant \frac{1}{C} \|\mathbf{x} - \mathbf{y}\|_2$ for any $t$ and some $C > 0$. Denote by $\mathbf{x}_{t_i}^{gt} = \mathbf{x}_{t_i}^{gt}(\widetilde{\mathbf{x}}_{t_K})$ the groundtruth intermediate result starting from $\widetilde{\mathbf{x}}_{t_K}$. Then we have the following inequality:*

$$\mathbb{E}_{\mathbf{x}_0, \boldsymbol{\epsilon}}[\|\widetilde{\mathbf{x}}_{t_{i-1}} - \mathbf{x}_{t_{i-1}}^{gt}\|_2] \leqslant C\Big( \sum_{n=i}^K \mathcal{L}_n(\tau_n)^{\frac{1}{2}} + \sum_{l=i}^K \mathbb{E}[\|\boldsymbol{\epsilon}_\theta(\mathbf{x}_{t_l}^{gt}, t_l) - \boldsymbol{\epsilon}_\theta(\mathbf{x}_{t_{l-1}}^{gt}, t_{l-1})\|_2] \Big). \tag{7}$$

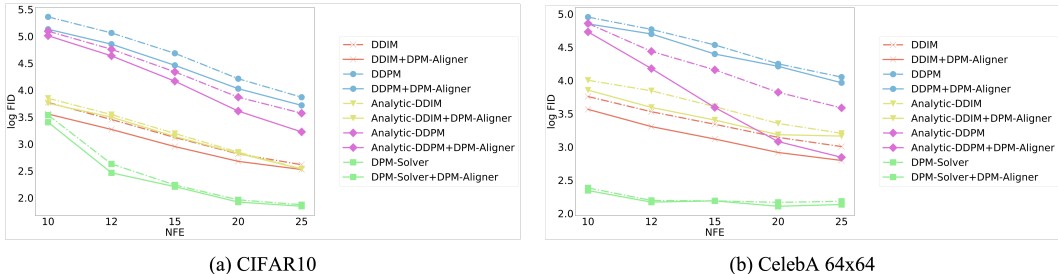

(a) CIFAR10      (b) CelebA 64x64

Figure 3: **Quantitative comparison** measured by log FID ↓ on CIFAR10 (Krizhevsky & Hinton, 2009) and CelebA 64x64 (Liu et al., 2015), under original DDPM framework. All are evaluated with different NFEs on the horizontal axis. We apply the quadratic trajectory for DDIM and DDPM, uniform trajectory for Analytic-DDIM and Analytic-DDPM, while we use log-SNR trajectory for DPM-Solver-2 on both datasets.

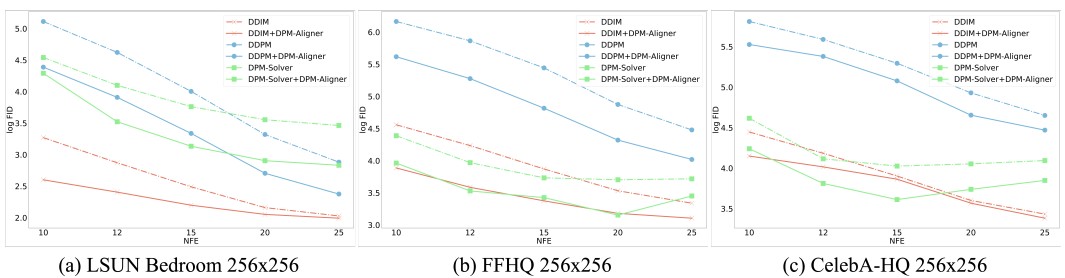

(a) LSUN Bedroom 256x256      (b) FFHQ 256x256      (c) CelebA-HQ 256x256

Figure 4: **Quantitative comparison** measured by log FID ↓ on LSUN Bedroom 256x256 (Yu et al., 2015), FFHQ 256x256 (Karras et al., 2019), and CelebA-HQ 256x256 (Karras et al., 2018), under LDM framework. All are evaluated with different NFEs on the horizontal axis. We apply the uniform trajectory for DDIM and DDPM, and log-SNR trajectory for DPM-Solver-2 on all datasets.

### 3.5 TRAINING STRATEGY OF DPM-ALIGNER

Recall that we employ a *sequential strategy* to train each $\tau_i$ for $i$ from $K$ to 1, *i.e.*, each $\tau_i$ needs to be optimized sequentially. Also, one can train all $\tau_i$'s simultaneously with Eq. (8) by a *parallel strategy*:

$$\mathcal{L}_i^{parallel}(\tau_i) = \mathbb{E}_{\mathbf{x}_0, \boldsymbol{\epsilon}} \left[ \| \boldsymbol{\epsilon}_\theta(f_{\theta,\tau}(\mathbf{x}_{t_i}, \tau_i), t_{i-1}) - \boldsymbol{\epsilon}_\theta(\mathbf{x}_{t_i}, t_i) \|_2^2 \right], \tag{8}$$

where $\mathbf{x}_{t_i} = \alpha_{t_i} \mathbf{x}_0 + \sigma_{t_i} \boldsymbol{\epsilon}$ instead of the intermediate denoised result $\widetilde{\mathbf{x}}_{t_i}$ by aligned DE solver. The parallel strategy is feasible since $\mathbf{x}_{t_i}$ is independent with all $\tau_i$'s, and one can train $\tau_i$ simultaneously by sampling different $\mathbf{x}_{t_i}$ on different GPUs. Despite the extra acceleration of the optimization process, the parallel training strategy will subtly harm the generation performance, which can also be observed from from Tab. 2 in Sec. 4.3. This is because the achieved sub-optimal $\tau_K, \cdots, \tau_{i+1}$ fail to ensure that the sampling distribution of $\widetilde{\mathbf{x}}_{t_i}$ matches the real distribution $q_{t_i}(\mathbf{x}_{t_i})$. Then $f_{\theta,\tau}(\widetilde{\mathbf{x}}_{t_i}, \tau_i)$ and $f_{\theta,\tau}(\mathbf{x}_{t_i}, \tau_i)$ have different distributions. Hence in order to make sure that $\widetilde{\mathbf{x}}_{t_{i-1}} = f_{\theta,\tau}(\widetilde{\mathbf{x}}_{t_i})$ follows $q_{t_{i-1}}(\mathbf{x}_{t_{i-1}})$, one can only use the biased $\widetilde{\mathbf{x}}_{t_i}$ for subsequent training of $\tau_i$. In other words, the parallel training strategy will introduce extra error at each denoising step.

## 4 EXPERIMENTS

In this section, we show that the proposed DPM-Aligner can greatly improve the performance of the baseline samplers of existing pre-trained DPM acceleration algorithms. We vary different number of funtion evaluations which is the number of calling the noise prediction model $\boldsymbol{\epsilon}_\theta$. We show the great improvement of sampling quality in Sec. 4.2, and two different training strategies in Sec. 4.3.

### 4.1 EXPERIMENTAL SETUPS

**Datasets and baselines.** We apply our DPM-Aligner to existing fast sampling methods by prior work, including DDIM (Song et al., 2021), DDPM (Ho et al., 2020), Analytic-DDIM (Bao

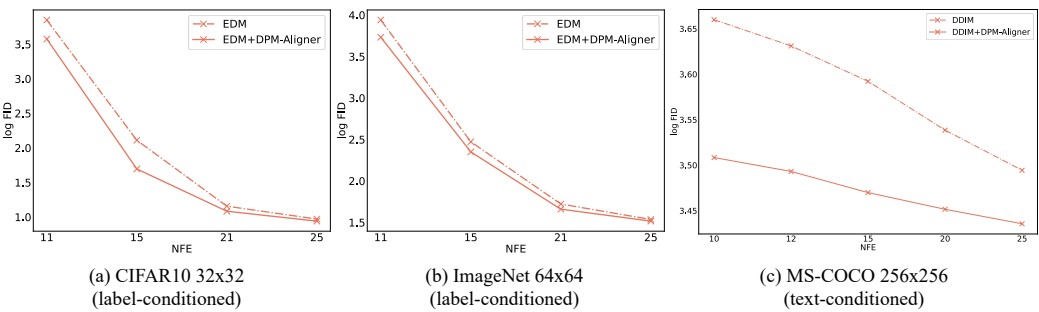

Figure 5: **Quantitative comparison** measured by log FID ↓ on CIFAR10 (Krizhevsky & Hinton, 2009), ImageNet 64x64 (Deng et al., 2009), and MS-COCO 256x256 (Lin et al., 2014), under EDM and LDM framework on conditional generation task. All are evaluated with different NFEs on the horizontal axis. We apply the orignally designed trajectory for EDM and linear trajectory for LDM.

et al., 2022), Analytic-DDPM (Bao et al., 2022) and DPM-Solver (Lu et al., 2022). We also apply `DPM-Aligner` on EDM (Karras et al., 2022), which is a high-order DE solver with specially designed noise schedule on CIFAR10 (Krizhevsky & Hinton, 2009) and ImageNet 64x64 (Deng et al., 2009). For high-resolution DPMs, we introduce the latent diffusion models (LDMs) (Rombach et al., 2022) as the DPM framework. Particularly, as for DPM-Solver, we employ the 2-order DE sampler, namely, DPM-Solver-2 instead of the fast version of DPM-Solver proposed by Lu *et al.* (Lu et al., 2022). The pre-trained DPMs are trained on CIFAR10 (Krizhevsky & Hinton, 2009), CelebA 64x64 (Liu et al., 2015), LSUN Bedroom 256x256 (Yu et al., 2015), FFHQ 256x256 (Karras et al., 2019), CelebA-HQ 256x256 (Karras et al., 2018), ImageNet 256x256 (Deng et al., 2009), and MS-COCO 256x256 (Lin et al., 2014), respectively. It is noteworthy that both EDM on ImageNet 64x64 and LDM on ImageNet 256x256 and MS-COCO 256x256 are conditional generation, which base on label, label, and text, respectively. DPMs on all seven datasets are trained with linear noise schedule (Ho et al., 2020). The number of total timesteps $T$ is 1000 for all seven datasets.

**Evaluation metrics.** We draw 50,000 samples and use Fréchet Inception Distance (FID) (Heusel et al., 2017) to evaluate the fidelity of the synthesized images. Inception Score (IS) (Salimans et al., 2016) measures how well a model captures the full ImageNet class distribution while still producing individual samples of a single class convincingly. To better measure spatial relationships, we introduce sFID (Nash et al., 2021), rewarding image distributions with coherent high-level structure. Finally, we use Improved Precision and Recall (Kynkäänniemi et al., 2019) to separately measure sample fidelity (precision) and diversity (recall).

**Implementation details.** We train the proposed `DPM-Aligner` on the platform of PyTorch (Paszke et al., 2019), in a Linux environment with an NVIDIA Tesla A100 GPU. We use the pre-trained model of CelebA 64x64 provided in the official implementation (Song et al., 2021), while the pre-trained models on CIFAR10 is trained by Bao *et al.* (Bao et al., 2022) with the same U-Net structure as Nichol & Dhariwal (Nichol & Dhariwal, 2021). For EDMs, we directly use the pre-trained model of CIFAR10 and ImageNet 64x64 provided in the official implementation. As for LDMs, we use the pre-trained model of LSUN Bedroom 256x256, FFHQ 256x256, CelebA-HQ 256x256, ImageNet 256x256, and MS-COCO 256x256 provided in the official implementation (Rombach et al., 2022).

### 4.2 SAMPLE QUALITY

**Unconditional generation on CIFAR10 and CelebA.** For the strongest baseline, we apply the quadratic trajectory for DDIM and DDPM on both CIFAR10 and CelebA 64x64 datasets, which empirically achieves better FID performance than that under uniform trajectory. As for DPM-Solver-2, we use the log-SNR trajectory following the setup of Lu *et al.* (Lu et al., 2022). As shown in Fig. 3 with the original DDPM framework, under all trajectories of different number of function evaluations (NFE) $K$, our proposed `DPM-Aligner` consistently improves the sampling performance of the original DDIM, DDPM, Analytic-DDIM, Analytic-DDPM, and DPM-Solver-2 significantly.

**Unconditional generation on LSUN Bedroom, FFHQ, and CelebA-HQ.** As for the datasets with high resolution 256x256, we apply the LDM (Rombach et al., 2022) framework to guarantee the

Table 1: **Quantitative comparison** measured by IS ↑, FID ↓, sFID ↓, Precision ↑ and Recall ↑ on LSUN Bedroom 256, FFHQ 256, CelebA-HQ 256, and ImageNet 256, respectively. All are evaluated by drawing 50,000 samples via DDIM sampler upon LDM, with 10 function evaluations (NFE = 10).

| LSUN Bedroom 256x256, *unconditional* generation | | | | | |
|---|---|---|---|---|---|
| Method | IS ↑ | FID ↓ | sFID ↓ | Precision ↑ | Recall ↑ |
| DDIM | 2.30 | 9.46 | 12.02 | 0.55 | 0.34 |
| DDIM + Ours | **2.31** | **5.85** | **9.44** | **0.57** | **0.44** |
| FFHQ 256x256, *unconditional* generation | | | | | |
| Method | IS ↑ | FID ↓ | sFID ↓ | Precision ↑ | Recall ↑ |
| DDIM | 4.00 | 23.58 | 14.59 | 0.63 | 0.21 |
| DDIM + Ours | **4.40** | **14.80** | **9.69** | **0.67** | **0.32** |
| CelebA-HQ 256x256, *unconditional* generation | | | | | |
| Method | IS ↑ | FID ↓ | sFID ↓ | Precision ↑ | Recall ↑ |
| DDIM | 2.95 | 18.72 | 16.68 | **0.68** | 0.19 |
| DDIM + Ours | **3.20** | **16.59** | **15.61** | 0.67 | **0.26** |
| ImageNet 256x256, *conditional* generation | | | | | |
| Method | IS ↑ | FID ↓ | sFID ↓ | Precision ↑ | Recall ↑ |
| DDIM | 324.52 | 10.13 | 12.52 | 0.91 | 0.28 |
| DDIM + Ours | **336.94** | **9.63** | **7.29** | **0.92** | **0.30** |

| (a)DDIM sampler, NFE = 10 uniform trajectory | (b) DDIM sampler, NFE = 15 uniform trajectory | (c) DDPM sampler, NFE = 10 uniform trajectory | (d) DPM-Solver-2 sampler, NFE = 10 log-SNR trajectory |
|---|---|---|---|

Figure 6: **Quantitative measurement** of the drop of the sampling quality by DDIM, DDPM, and DPM-Sovler under different types of trajectories. The horizontal axis represents the number of re-aligned timesteps from $t_K$ down to $t_1$ with $K = 10$ or 15, respectively. The red line shows the FID drop, while the gray dashed line shows the FID of the baseline sampler.

training efficiency of our algorithm without loss of the synthesis performance. From Fig. 4 one can conclude that our algorithm achieves even better performance improvement on the high-resolution datasets. To quantitatively demonstrate the performance improvement of our method, we make further comparison using more metrics, as shown in Tab. 1. And the qualitative results can be found in Fig. 7, clearly demonstrating the quality improvement.

**High-order sampler generation using EDM.** As for high-order DE solver, we apply our method on EDM (Karras et al., 2022), which introduces the 2nd Heun sampling method on label-conditioned generation. As demonstrated in Fig. 5, our method improves the generation performance as well, indicating the great compatibility for the conditional generation tasks under high-order DE solvers.

**Label-conditioned generation on ImageNet.** As shown in Tab. 1 and Fig. 7, the synthesis performance of our algorithm surpasses the baseline significantly both qualitatively and quantitatively.

**Text-conditioned generation on MS-COCO.** As for the most challenging text-conditioned generation task, the qualitative and quantitative results demonstrated in Fig. 7, Fig. 5, and Tab. 1 confirm the compatibility and capability of our method.

It is noteworthy that the sampling quality improvement is more significant with a small NFE. For instance, the improvement of FID between DDIM and DDIM+Ours on FFHQ decreases from 8.77 to 1.51 as NFE grows from 10 to 25. This roots in the fact that the larger the NFE is, the smaller the truncation error is, and hence the smaller the gap between the real and sampling distributions is.

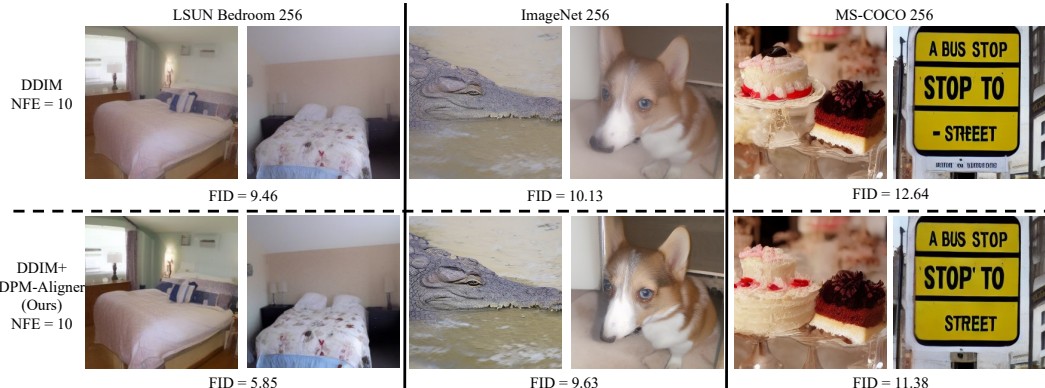

Figure 7: **Comprehensive comparison** on LSUN Bedroom 256x256 (Yu et al., 2015), ImageNet 256x256 (Deng et al., 2009), and MS-COCO 256x256 (Lin et al., 2014), under original LDM framework. The three tasks are unconditional, label-conditioned, and text-conditioned generation, respectively. All are evaluated with 10 NFEs with uniform trajectory and DDIM sampler.

Table 2: **Quantitative comparison** measured by FID ↓ between *sequential* and *parallel* training strategies of the proposed DPM-Aligner, where all are evaluated by drawing 50,000 samples via DDIM (Song et al., 2021). We conduct experiments on 5 datasets, *i.e.*, CIFAR10 (Krizhevsky & Hinton, 2009), CelebA 64x64 (Liu et al., 2015), LSUN Bedroom 256x256 (Yu et al., 2015), FFHQ 256x256 (Karras et al., 2019), and CelebA-HQ 256x256 (Karras et al., 2018). We apply the quadratic trajectory for CIFAR10 and CelebA 64x64 datasets, and uniform trajectory on the other three datasets.

| Dataset | CIFAR10 | CelebA | LSUN Bedroom | FFHQ | CelebA-HQ |
|---|---|---|---|---|---|
| DDIM | 13.65 | 13.53 | 9.65 | 23.57 | 21.81 |
| DDIM + Ours (sequential) | 11.77 | **11.84** | **6.07** | **14.80** | **17.75** |
| DDIM + Ours (parallel) | **11.75** | 12.14 | 6.08 | 15.03 | 18.03 |

We also confirm the performance improvement by re-aligning the timesteps one by one. As shown in Fig. 6, by re-aligning the timestep gradually, the FID decreases monotonically, which confirms the correctness of Theorem 2 and the effectiveness of the proposed method. One can also verify the effectiveness of the proposed DPM-Aligner from Fig. 2. By applying DPM-Aligner to re-align the timestep $t_i$ gradually, the gap between the sampling and real distribution decreases significantly at each step for all timestep trajectories, NFEs, and DPM samplers, which demonstrates the strong capability of correcting the one-step truncation error, and hence the accumulative truncation error.

### 4.3 TRAINING STRATEGY OF DPM-ALIGNER

Recall that we separately introduce a *sequential strategy* and a *parallel strategy* to train each timestep $\tau_i$, for $i$ from $K$ to 1 in Sec. 3.5. We deduce that there is a performance gap between the two strategies, mainly due to the extra error introduced by the parallel strategy at each denoising step. This can also be observed and concluded from Tab. 2 clearly. Nevertheless, the parallel training strategy achieves on-par sampling performance, which provides a far more efficient version of DPM-Aligner empirically, and is extremely significant to train DPM-Aligner of large NFE cases. Therefore, how to improve the performance of the parallel training strategy will be an interesting avenue for future research.

## 5 CONCLUSION

In this paper, we propose a plug-in algorithm for more accurate diffusion model acceleration, which replaces the original timestep to a re-aligned one. We provide a proof to show the feasibility to achieve a better sampling performance by simply re-aligning the timestep. We also give an estimation of error bound for the deterministic DE solver theoretically. We conduct comprehensive experiments to demonstrate significant improvement of sampling quality under different NFEs.

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

# A PROOFS AND DERIVATIONS

In this section, we will prove the theorems claimed in the main manuscript.

## A.1 PROOF OF THEOREM 1

**Theorem 1.** *Assume that $\epsilon_\theta$ is the groundtruth noise prediction model. Then the training process of* DPM-Aligner *resembles that of the original DPM, i.e., for $i = K, K - 1, \cdots, 1$, the optimal $\tau_i = \arg\min_{\tau_i} \mathcal{L}_i(\tau_i)$ holds the following property:*

$$\arg\min_{\tau_i} \mathcal{L}_i(\tau_i) = \arg\min_{\tau_i} \mathbb{E}_{\mathbf{x}_0, \epsilon} \left[ \|\epsilon_\theta(f_{\theta,\tau}(\widetilde{\mathbf{x}}_{t_i}, \tau_i), t_{i-1}) - \frac{\widetilde{\mathbf{x}}_{t_i} - \alpha_{t_i}\mathbf{x}_0}{\sigma_{t_i}}\|_2^2 \right]. \tag{1}$$

We first claim the following lemmas which are crucial for the proof of Theorem 1.

**Lemma 1.** *Let $\mathbf{x}_0 \sim q_0(\mathbf{x}_0)$, and $q_{0t}(\mathbf{x}_t|\mathbf{x}_0) = \mathcal{N}(\mathbf{x}_t; \alpha_t\mathbf{x}_0, \sigma_t^2\mathbf{I})$. Denote by $q_t(\mathbf{x}_t)$ the marginal distribution of $\mathbf{x}_t$. Then we have $\nabla \log q_t(\mathbf{x}_t) = -\mathbb{E}\left[ \frac{\mathbf{x}_t - \alpha_t\mathbf{x}_0}{\sigma_t^2} | \mathbf{x}_t \right]$.*

*Proof of Lemma 1.* According to the definition of $q_t(\mathbf{x}_t)$, one can notice that $\nabla \log q_t(\mathbf{x}_t) = \nabla_{\mathbf{x}_t} \log \int q_0(\mathbf{x}_0) q_{0t}(\mathbf{x}_t|\mathbf{x}_0)d\mathbf{x}_0$. Then we have

$$\nabla \log q_t(\mathbf{x}_t) = \frac{\int q_0(\mathbf{x}_0) \nabla_{\mathbf{x}_t} q_{0t}(\mathbf{x}_t|\mathbf{x}_0)d\mathbf{x}_0}{\int q_0(\mathbf{x}_0) q_{0t}(\mathbf{x}_t|\mathbf{x}_0)d\mathbf{x}_0} \tag{2}$$

$$= \frac{\int q_0(\mathbf{x}_0) q_{0t}(\mathbf{x}_t|\mathbf{x}_0) \nabla_{\mathbf{x}_t} \log q_{0t}(\mathbf{x}_t|\mathbf{x}_0)d\mathbf{x}_0}{q_t(\mathbf{x}_t)} \tag{3}$$

$$= \int \frac{q_0(\mathbf{x}_0) q_{0t}(\mathbf{x}_t|\mathbf{x}_0)}{q_t(\mathbf{x}_t)} \nabla_{\mathbf{x}_t} \log q_{0t}(\mathbf{x}_t|\mathbf{x}_0)d\mathbf{x}_0 \tag{4}$$

$$= \int q(\mathbf{x}_0|\mathbf{x}_t) \nabla_{\mathbf{x}_t} \log q_{0t}(\mathbf{x}_t|\mathbf{x}_0)d\mathbf{x}_0 \tag{5}$$

$$= \mathbb{E}[\nabla_{\mathbf{x}_t} \log q_{0t}(\mathbf{x}_t|\mathbf{x}_0)|\mathbf{x}_t] \tag{6}$$

$$= -\mathbb{E}\left[ \frac{\mathbf{x}_t - \alpha_t\mathbf{x}_0}{\sigma_t^2} | \mathbf{x}_t \right], \tag{7}$$

where Eq. (5) comes from Bayes' rule. $\square$

**Lemma 2.** *Let $g(\mathbf{x}_t), h(\mathbf{x}_t, \mathbf{x}_0)$ be integrable functions, then the following equality holds.*

$$\mathbb{E}_{q(\mathbf{x}_t)}[\langle g(\mathbf{x}_t), \mathbb{E}_{q(\mathbf{x}_0|\mathbf{x}_t)}[h(\mathbf{x}_t, \mathbf{x}_0)|\mathbf{x}_t]\rangle] = \mathbb{E}_{q(\mathbf{x}_t)}[\langle g(\mathbf{x}_t), h(\mathbf{x}_t, \mathbf{x}_0)\rangle]. \tag{8}$$

*Proof of Lemma 2.* Note that

$$\mathbb{E}_{q(\mathbf{x}_t)}[\langle g(\mathbf{x}_t), \mathbb{E}_{q(\mathbf{x}_0|\mathbf{x}_t)}[h(\mathbf{x}_t, \mathbf{x}_0)|\mathbf{x}_t]\rangle] = \int \langle g(\mathbf{x}_t), \mathbb{E}_{q(\mathbf{x}_0|\mathbf{x}_t)}[h(\mathbf{x}_t, \mathbf{x}_0)|\mathbf{x}_t]\rangle p(\mathbf{x}_t)d\mathbf{x}_t \tag{9}$$

$$= \int \langle g(\mathbf{x}_t), \int h(\mathbf{x}_t, \mathbf{x}_0)p(\mathbf{x}_0|\mathbf{x}_t)d\mathbf{x}_0\rangle p(\mathbf{x}_t)d\mathbf{x}_t \tag{10}$$

$$= \int \int \langle g(\mathbf{x}_t), h(\mathbf{x}_t, \mathbf{x}_0)\rangle p(\mathbf{x}_0|\mathbf{x}_t)p(\mathbf{x}_t)d\mathbf{x}_0 d\mathbf{x}_t \tag{11}$$

$$= \int \int \langle g(\mathbf{x}_t), h(\mathbf{x}_t, \mathbf{x}_0)\rangle p(\mathbf{x}_0, \mathbf{x}_t)d\mathbf{x}_0 d\mathbf{x}_t \tag{12}$$

$$= E_{q(\mathbf{x}_t)}[\langle g(\mathbf{x}_t), h(\mathbf{x}_t, \mathbf{x}_0)\rangle]. \tag{13}$$

where Eq. (11) is by linearity of integral. $\square$

Then we start to prove the Theorem 1 as below.

*Proof of Theorem 1.* Given the assumption that $\epsilon_\theta$ is the groundtruth noise prediction model, we have $\epsilon_\theta(\mathbf{x}_t, t) = \mathbb{E}[\frac{\mathbf{x}_t - \alpha_t \mathbf{x}_0}{\sigma_t} | \mathbf{x}_t]$ from Lemma 1. Then we have

$$\mathcal{L}_i(\tau_i) = \mathbb{E}_{\mathbf{x}_0, \epsilon} \left[ \|\epsilon_\theta(f_{\theta,\tau}(\widetilde{\mathbf{x}}_{t_i}, \tau_i), t_{i-1}) - \epsilon_\theta(\widetilde{\mathbf{x}}_{t_i}, t_i)\|_2^2 \right] \tag{14}$$

$$= \mathbb{E}_{\mathbf{x}_0, \epsilon} \left[ \|\epsilon_\theta(f_{\theta,\tau}(\widetilde{\mathbf{x}}_{t_i}, \tau_i), t_{i-1}) i\|_2^2 + \|\epsilon_\theta(\widetilde{\mathbf{x}}_{t_i}, t_i)\|_2^2 \right]$$
$$- 2\mathbb{E}_{\mathbf{x}_0, \epsilon} \left[ \langle \epsilon_\theta(f_{\theta,\tau}(\widetilde{\mathbf{x}}_{t_i}, \tau_i), t_{i-1}), \epsilon_\theta(\widetilde{\mathbf{x}}_{t_i}, t_i) - \epsilon \rangle \right] \tag{15}$$

$$= \mathbb{E}_{\mathbf{x}_0, \epsilon} \left[ \|\epsilon_\theta(f_{\theta,\tau}(\widetilde{\mathbf{x}}_{t_i}, \tau_i), t_{i-1}) i\|_2^2 + \|\epsilon_\theta(\widetilde{\mathbf{x}}_{t_i}, t_i)\|_2^2 \right]$$
$$- 2\mathbb{E}_{\mathbf{x}_0, \epsilon} \left[ \left\langle \epsilon_\theta(f_{\theta,\tau}(\widetilde{\mathbf{x}}_{t_i}, \tau_i), t_{i-1}), \mathbb{E} \left[ \frac{\widetilde{\mathbf{x}}_{t_i} - \alpha_{t_i} \mathbf{x}_0}{\sigma_{t_i}} | \widetilde{\mathbf{x}}_{t_i} \right] \right\rangle \right] \tag{16}$$

$$= \mathbb{E}_{\mathbf{x}_0, \epsilon} \left[ \|\epsilon_\theta(f_{\theta,\tau}(\widetilde{\mathbf{x}}_{t_i}, \tau_i), t_{i-1})\|_2^2 + \|\epsilon_\theta(\widetilde{\mathbf{x}}_{t_i}, t_i)\|_2^2 \right]$$
$$- 2\mathbb{E}_{\mathbf{x}_0, \epsilon} \left[ \left\langle \epsilon_\theta(f_{\theta,\tau}(\widetilde{\mathbf{x}}_{t_i}, \tau_i), t_{i-1}), \frac{\widetilde{\mathbf{x}}_{t_i} - \alpha_{t_i} \mathbf{x}_0}{\sigma_{t_i}} \right\rangle \right] \tag{17}$$

$$= \mathbb{E}_{\mathbf{x}_0, \epsilon} \left[ \|\epsilon_\theta(f_{\theta,\tau}(\widetilde{\mathbf{x}}_{t_i}, \tau_i), t_{i-1}) - \frac{\widetilde{\mathbf{x}}_{t_i} - \alpha_{t_i} \mathbf{x}_0}{\sigma_{t_i}}\|_2^2 \right]$$
$$, \qquad + \mathbb{E}_{\mathbf{x}_0, \epsilon} \left[ \|\epsilon_\theta(\widetilde{\mathbf{x}}_{t_i}, t_i)\|_2^2 - \|\frac{\widetilde{\mathbf{x}}_{t_i} - \alpha_{t_i} \mathbf{x}_0}{\sigma_{t_i}}\|_2^2 \right], \tag{18}$$

where Eq. (17) is due to Lemma 2. Since $\|\epsilon_\theta(\widetilde{\mathbf{x}}_{t_i}, t_i)\|_2^2 - \|\frac{\widetilde{\mathbf{x}}_{t_i} - \alpha_{t_i} \mathbf{x}_0}{\sigma_{t_i}}\|_2^2$ is independent with $\tau_i$, we have

$$\arg\min_{\tau_i} \mathcal{L}_i(\tau_i) = \arg\min_{\tau_i} \mathbb{E}_{\mathbf{x}_0, \epsilon} \left[ \|\epsilon_\theta(f_{\theta,\tau}(\widetilde{\mathbf{x}}_{t_i}, \tau_i), t_{i-1}) - \frac{\widetilde{\mathbf{x}}_{t_i} - \alpha_{t_i} \mathbf{x}_0}{\sigma_{t_i}}\|_2^2 \right]. \tag{19}$$

$\square$

**Remark 1.** *Note that the objective of the original DPM has the following form:*

$$\mathbb{E}_{\mathbf{x}_0, \epsilon} \left[ \|\epsilon_\theta(\mathbf{x}_{t_i}, t_i) - \epsilon\|_2^2 \right] = \mathbb{E}_{\mathbf{x}_0, \epsilon} \left[ \|\epsilon_\theta(\mathbf{x}_{t_i}, t_i) - \frac{\mathbf{x}_{t_i} - \alpha_{t_i} \mathbf{x}_0}{\sigma_{t_i}}\|_2^2 \right], \tag{20}$$

*which has a similar form as the objective in Theorem 1.*

### A.2 PROOF OF THEOREM 2

**Theorem 2.** *Under the condition in Theorem 1, and let $f_{\theta,\tau}$ be a deterministic sampler. Assume that $\|\epsilon_\theta(\mathbf{x}, t) - \epsilon_\theta(\mathbf{y}, t)\|_2 \geq \frac{1}{C} \|\mathbf{x} - \mathbf{y}\|_2$ for any $t$ and some $C > 0$. Denote by $\mathbf{x}_{t_i}^{gt} = \mathbf{x}_{t_i}^{gt}(\widetilde{\mathbf{x}}_{t_K})$ the groundtruth intermediate result starting from $\widetilde{\mathbf{x}}_{t_K}$. Then we have the following inequality:*

$$\mathbb{E}_{\mathbf{x}_0, \epsilon}[\|\widetilde{\mathbf{x}}_{t_{i-1}} - \mathbf{x}_{t_{i-1}}^{gt}\|_2] \leqslant C \left( \sum_{n=i}^K \mathcal{L}_n(\tau_n)^{\frac{1}{2}} + \sum_{l=i}^K \mathbb{E}[\|\epsilon_\theta(\mathbf{x}_{t_l}^{gt}, t_l) - \epsilon_\theta(\mathbf{x}_{t_{l-1}}^{gt}, t_{l-1})\|_2] \right). \tag{21}$$

*Proof of Theorem 2.* By the assumption, we have

$$\mathbb{E}_{\mathbf{x}_0, \epsilon}[\|\widetilde{\mathbf{x}}_{t_{i-1}} - \mathbf{x}_{t_{i-1}}^{gt}\|_2] \leqslant C \mathbb{E}_{\mathbf{x}_0, \epsilon}[\|\epsilon_\theta(f_{\theta,\tau}(\widetilde{\mathbf{x}}_{t_i}, \tau_i), t_{i-1}) - \epsilon_\theta(\mathbf{x}_{t_{i-1}}^{gt}, t_{i-1})\|_2] \tag{22}$$

Define $e_{i-1} = \epsilon_\theta(f_{\theta,\tau}(\widetilde{\mathbf{x}}_{t_i}, \tau_i), t_{i-1}) - \epsilon_\theta(\mathbf{x}_{t_{i-1}}^{gt}, t_{i-1})$. Then we can easily derive that

$$e_{i-1} = \epsilon_\theta(f_{\theta,\tau}(\widetilde{\mathbf{x}}_{t_i}, \tau_i), t_{i-1}) - \epsilon_\theta(\widetilde{\mathbf{x}}_{t_i}, t_i) + \epsilon_\theta(\widetilde{\mathbf{x}}_{t_i}, t_i) - \epsilon_\theta(\mathbf{x}_{t_i}^{gt}, t_i)$$
$$+ \epsilon_\theta(\mathbf{x}_{t_i}^{gt}, t_i) - \epsilon_\theta(\mathbf{x}_{t_{i-1}}^{gt}, t_{i-1}) \tag{23}$$

$$= \epsilon_\theta(f_{\theta,\tau}(\widetilde{\mathbf{x}}_{t_i}, \tau_i), t_{i-1}) - \epsilon_\theta(\widetilde{\mathbf{x}}_{t_i}, t_i) + e_i + \epsilon_\theta(\mathbf{x}_{t_i}^{gt}, t_i) - \epsilon_\theta(\mathbf{x}_{t_{i-1}}^{gt}, t_{i-1}) \tag{24}$$

$$= \sum_{n=i}^{K-1} \left( \epsilon_\theta(f_{\theta,\tau}(\widetilde{\mathbf{x}}_{t_n}, \tau_n), t_{n-1}) - \epsilon_\theta(\widetilde{\mathbf{x}}_{t_n}, t_n) \right) + e_{K-1}$$
$$+ \sum_{l=i}^{K-1} \left( \epsilon_\theta(\mathbf{x}_{t_l}^{gt}, t_l) - \epsilon_\theta(\mathbf{x}_{t_{l-1}}^{gt}, t_{l-1}) \right), \tag{25}$$

where Eq. (24) is due to $\widetilde{\mathbf{x}}_{t_i} = f_{\theta,\tau}(\widetilde{\mathbf{x}}_{t_{i+1}}, \tau_{i+1})$. Since $\mathbf{x}^{gt}_{t_K} = \widetilde{\mathbf{x}}_{t_K}$, we have

$$e_{K-1} = \boldsymbol{\epsilon}_\theta(f_{\theta,\tau}(\widetilde{\mathbf{x}}_{t_K}, \tau_K), t_{K-1}) - \boldsymbol{\epsilon}_\theta(\widetilde{\mathbf{x}}_{t_K}, t_K) + \boldsymbol{\epsilon}_\theta(\mathbf{x}^{gt}_{t_K}, t_K) - \boldsymbol{\epsilon}_\theta(\mathbf{x}^{gt}_{t_{K-1}}, t_{K-1}). \tag{26}$$

Then we have

$$e_{i-1} = \sum_{n=i}^{K} \left( \boldsymbol{\epsilon}_\theta(f_{\theta,\tau}(\widetilde{\mathbf{x}}_{t_n}, \tau_n), t_{n-1}) - \boldsymbol{\epsilon}_\theta(\widetilde{\mathbf{x}}_{t_n}, t_n) \right) + \sum_{l=i}^{K} \left( \boldsymbol{\epsilon}_\theta(\mathbf{x}^{gt}_{t_l}, t_l) - \boldsymbol{\epsilon}_\theta(\mathbf{x}^{gt}_{t_{l-1}}, t_{l-1}) \right), \tag{27}$$

and

$$\mathbb{E}[\|\boldsymbol{\epsilon}_\theta(f_{\theta,\tau}(\widetilde{\mathbf{x}}_{t_i}, \tau_i), t_{i-1}) - \boldsymbol{\epsilon}_\theta(\mathbf{x}^{gt}_{t_{i-1}}, t_{i-1})\|_2] \tag{28}$$

$$\leqslant \sum_{n=i}^{K} \mathbb{E}[\|\boldsymbol{\epsilon}_\theta(f_{\theta,\tau}(\widetilde{\mathbf{x}}_{t_n}, \tau_n), t_{n-1}) - \boldsymbol{\epsilon}_\theta(\widetilde{\mathbf{x}}_{t_n}, t_n)\|_2]$$

$$+ \sum_{l=i}^{K} \mathbb{E}[\|\boldsymbol{\epsilon}_\theta(\mathbf{x}^{gt}_{t_l}, t_l) - \boldsymbol{\epsilon}_\theta(\mathbf{x}^{gt}_{t_{l-1}}, t_{l-1})\|_2] \tag{29}$$

$$\leqslant \sum_{n=i}^{K} \mathcal{L}_n(\tau_n)^{\frac{1}{2}} + \sum_{l=i}^{K} \mathbb{E}[\|\boldsymbol{\epsilon}_\theta(\mathbf{x}^{gt}_{t_l}, t_l) - \boldsymbol{\epsilon}_\theta(\mathbf{x}^{gt}_{t_{l-1}}, t_{l-1})\|_2], \tag{30}$$

where Eq. (30) is due to Cauchy inequality. Combine Eq. (22) and Eq. (30), we prove the theorem. $\square$

## B  PSEUDO-CODE OF TRAINING PROCESS

Recall that we introduce two different training strategies for the proposed `DPM-Aligner`, *i.e.*, the *sequenatial strategy* and the *parallel strategy*. We have proved the equivalence of the two training strategies, and analyzed the performance difference between the two strategies upon DDIM (Song et al., 2021). In this part, we provide the pseudo-codes of the two training strategies in Algorithm 1 and Algorithm 2.

**Algorithm 1** Pseudo-code of sequential training strategy of `DPM-Aligner` in a PyTorch-like style.

```
1  import torch
2
3
4  def sequential_training_loss(x_0, t_list, tau_list, i, tau_i, F, E):
5      """Defines the forward process of one sequential training step.
6
7      Args:
8          x_0: Data inputs, with shape [B, C, H, W].
9          t_list: The preset timestep trajectory from 0 to T.
10         tau_list: The list consist of previously achieved re-aligned timesteps from tau_K to
       tau_ip1.
11         i: The index of current timestep tau.
12         tau_i: The timestep to re-align.
13         F: The DE solver to denoise the input 'x' from timestep 't' to timestep 's' using re-
       aligned input condition 'tau'.
14         E: The noise prediction model with input 'x' and 't'.
15     """
16     # Compute the x_T at timestep T.
17     z_T = torch.randn_like(x_0)
18     x_T = alpha_T * x_0 + sigma_T * z_T
19
20     # Compute the denoised intermediate x_t_i
21     x = x_T
22     for tau, t, t_prev in zip(tau_list, t_list[::-1], t_list[-2::-1]):
23         x = F(x, t, t_prev, tau)
24     x_t_i = x
25
26     # Get the current and the previous timestep.
27     t_i, t_im1 = t_list[i], t_list[i - 1]
28
29     # Compute the denoised intermediate x_t_im1 with tau_i
30     x_t_im1 = F(x_t_i, t_i, t_im1, tau_i)
31
32     # Learn the translator.
33     loss = (E(x_t_im1, t_im1) - E(x_t_i, t_i)).square().mean()
34
35     return loss
```

**Algorithm 2** Pseudo-code of parallel training strategy of `DPM-Aligner` in a PyTorch-like style.

```
1  import torch
2
3
4  def sequential_training_loss(x_0, t_list, i, tau_i, F, E):
5      """Defines the forward process of one parallel training step.
6
7      Args:
8          x_0: Data inputs, with shape [B, C, H, W].
9          t_list: The preset timestep trajectory from 0 to T.
10         i: The index of current timestep tau.
11         tau_i: The timestep to re-align.
12         F: The DE solver to denoise the input 'x' from timestep 't' to timestep 's' using re-
       aligned input condition 'tau'.
13         E: The noise prediction model with input 'x' and 't'.
14     """
15     # Get the current and the previous timestep.
16     t_i, t_im1 = t_list[i], t_list[i - 1]
17
18     # Compute the x_t_i at timestep t_i.
19     z_t_i = torch.randn_like(x_0)
20     x_t_i = alpha_t_i * x_0 + sigma_t_i * z_t_i
21
22     # Compute the denoised intermediate x_t_im1 with tau_i
23     x_t_im1 = F(x_t_i, t_i, t_im1, tau_i)
24
25     # Learn the translator.
26     loss = (E(x_t_im1, t_im1) - E(x_t_i, t_i)).square().mean()
27
28     return loss
```

