# OpenReview forum: "Towards More Accurate Diffusion Model Acceleration with A Timestep Aligner"
_ICLR.cc/2024/Conference — ICLR 2024 Conference Withdrawn Submission_

### Official Review · Reviewer_koqD · 2023-10-18

**Soundness:** 2 fair
**Presentation:** 2 fair
**Contribution:** 1 poor
**Rating:** 3
**Confidence:** 4

**Summary:**

This work considers the acceleration of the sampling of diffusion models via a timestep aligner that finds a more accurate integral direction. The authors claim (in the Conclusion) to have provided theoretical guarantees showing the feasibility to achieve a better sampling performance by simply re-aligning the timestep. They also claim (in the abstract) that their extensive experimental results have shown that the re-aligned timesteps can help boost the inference performance of various SOTA acceleration methods.

**Strengths:**

The idea of re-aligning the timesteps may be of interest and the training of the re-aligned timesteps is not time-consuming.

**Weaknesses:**

1. There are serious overstated claims in this work:

- The authors mainly compare with DDIM, as well as DPM-Solver and Analytic-DDIM in certain cases. These methods are claimed to be SOTA acceleration methods in the abstract. However, this claim is not true. The acceleration of the sampling of diffusion models has been a popular topic, and there have been more powerful methods recently. In particular, the reported numerical results in this submission are fairly weak compared to those presented in recent works such as DPM-Solver++ and UniPC (the corresponding papers are publicly available for more than 6 months and the authors should not ignore these methods). For example, the authors highlighted in the abstract that "when using 10 denoising steps on the popular LSUN Bedroom dataset, we improve the FID of DDIM from 9.65 to 6.07". However, for 10-step denoising of LSUN Bedroom, the FID of DPM-Solver++/UniPC can be as low as 3.63/3.54 (see Table 1 of UniPC), and even DPM-Solver can achieve an FID of 6.10 (see Table 5 of DPM-Solver). Additionally, in some figures, the authors report the numerical results in terms of log FID, which is strange to me.

Lu, C., Zhou, Y., Bao, F., Chen, J., Li, C., & Zhu, J. (2022). Dpm-solver++: Fast solver for guided sampling of diffusion probabilistic models. arXiv preprint arXiv:2211.01095.

Zhao, W., Bai, L., Rao, Y., Zhou, J., & Lu, J. (2023). UniPC: A Unified Predictor-Corrector Framework for Fast Sampling of Diffusion Models. arXiv preprint arXiv:2302.04867.

- I cannot see how the theoretical results of this submission support the authors' claim in the Conclusion that "We provide a proof to show the feasibility to achieve a better sampling performance by simply re-aligning the timestep". In particular, without any competing theoretical results, how can the authors demonstrate "better sampling performance" through their Theorem 2? It appears that the findings in Theorem 2 are not directly related to the proposed timestep aligner. Additionally, the requirement of a lower bound of $\frac{1}{C}\\|\mathbf{x}-\mathbf{y}\\|_2$ seems uncommon in theoretical works on diffusion models, where it is more common to only have an upper bound. The authors should provide explanations for this seemingly strict assumption.

2. Some problems in the writing. For example,

- In the paragraph after Equation (4), the notation of $f_{\theta, \tau}(\tilde{\mathbf{x}}_{t_i}, \tau_i)$ is somewhat misleading:

$\tau_i$ only affects the evaluation of the neural function and do not affect the values of $\sigma$ or $\alpha$ (i.e., the authors still use $\sigma_{t_i}$ and $\alpha_{t_i}$, instead of $\sigma_{\tau_i}$ and $\alpha_{\tau_i}$).

- In Theorem 1, please revise $\tau_i = \arg\min_{\tau_i} \mathcal{L}_i(\tau_i)$.

**Questions:**

See Weaknesses.

---

### Official Review · Reviewer_n1ad · 2023-11-01

**Soundness:** 2 fair
**Presentation:** 3 good
**Contribution:** 2 fair
**Rating:** 3
**Confidence:** 4

**Summary:**

This paper proposes to accelerate the denoising process of the diffusion model. Given NFE with a large sampling interval, the authors aims to align the sampled distribution with to the real distribution at each step of NFE using a new timestep, which was then used to replace the original timestep in the denoising process. The experiments have been performed on various dataset with a consistent FID improvement.

**Strengths:**

1. The noise distribution alignment with a new step $\tau$ is efficient and easy to implement, which does not need to modify the parameters of the pretrained DPMs.
2. The learning process of \tau can be further accelerated by training all $\tau$'s in parallel with subtle result degradation.
3. The experiments exhibit a consistent FID improvement on various datasets.

**Weaknesses:**

1. The inequality assumption in Theorem 2 is too strong, which would not hold for deep neural networks due to the highly nonlinearity. This requires that there are no equipotential lines in the landscape of $\epsion_{\theta}$. It can be expected that the parameter $C$ in the inequality has to be a very large number  and even infinity, which makes the conclusion meaningless.
2. As the quoted intuitive in Paragraph 1 on Page 2, the authors propose to use the direction estimated from an intermediate state for the denoising process, in this way, does the magnitude of the direction, i.e., the coefficient of the noise in Eq. 4 affect to final results? If so, how sensitive the results are and how to estimate a better magnitude?
3. Apart from $\tau_i$, how does the current estimated distribution $\tilde{x}\_{t_i}$ affect the noise estimation in $\epsilon\_\theta(\tilde{x}\_{t_i}, \tau_i)$? It is unclear why one can use a fixed $\tau$ for generating all images.
4. I can understand that the key idea is to enforce the sampling distribution of intermediate $\tilde{x}\_{t_i}$ to coincide with the real distribution $q_{t_i}(x_{t_i})$, but why the target noise in Eq. 5 is $\epsilon\_\theta(\tilde{x}\_{t_i}, t_i)$? Does this noise still suffer from the truncation error?
5. In Fig. 2d, it seems the difference in the truncation errors for the proposed and the original methods is not as significant as other subfigures, does that mean for some advanced diffusion models (e.g., DPM-Solver-2), simply use a large sampling interval without alignment could also give a good result?

**Questions:**

1. Should $f\_\theta(\tilde{x}\_t, t)$ on the red line of Fig. 1a be $f_\theta(\tilde{x}\_{t-1}, t-1)$?
2. I suggest the authors put Fig. 1 and Fig. 2 onto the page where they are referred to avoid the readers searching them repeatedly.

---

### Official Review · Reviewer_j67R · 2023-11-01

**Soundness:** 3 good
**Presentation:** 3 good
**Contribution:** 2 fair
**Rating:** 5
**Confidence:** 2

**Summary:**

The work proposes to learn the timesteps for fast sampling techniques where the number of timesteps are often reduced. The main problem that the paper aims to is to reduce the truncation error resulting by timesteps skipping in the previous method. The timesteps are directly learned through gradient descent.

**Strengths:**

The paper presents an interesting method to improve the image quality generated by fast diffusion sampling techniques. This method might be useful for the communities.

**Weaknesses:**

Although the methods are interesting, there are some concerns:

1. The problem is not new and has been investigated before Wang (2023) and Watson(2021) The analysis in the paper has not been so comprehensive.
2. There should be some comparisons with Wang (2023) and Watson(2021)
3. Although the conceptual description is provided, it is necessary to prove that the proposed method actually helps to improve the truncation error as mentioned in the introduction.

Wang, Yunke, et al. "Learning to schedule in diffusion probabilistic models." Proceedings of the 29th ACM SIGKDD Conference on Knowledge Discovery and Data Mining. 2023.

Watson, Daniel, et al. "Learning to efficiently sample from diffusion probabilistic models." arXiv preprint arXiv:2106.03802 (2021).

**Questions:**

See the weaknesses

**Details Of Ethics Concerns:**

N.A